# Investigating the Key Factors Influencing the Process Innovation Capability in Organizations: Evidence from the Republic of Serbia

Marina Žižakov [ID], Teodora Vuckovic [ID], Srđan Vulanović *, Dušanka Dakić and Milan Delić [ID]

Faculty of Technical Sciences, University of Novi Sad, 21000 Novi Sad, Serbia; marinazizakov@uns.ac.rs (M.Ž.); teodora.lolic@uns.ac.rs (T.V.); dakic.dusanka@uns.ac.rs (D.D.); delic@uns.ac.rs (M.D.)
* Correspondence: srdjanv@uns.ac.rs

**Abstract:** Research exploring quality management, knowledge management, and innovations in organizations has received significant attention from academics worldwide, providing different insights. Innovation has been widely seen as an essential organizational performance driver. This study aims to accentuate the importance of quality management and knowledge management and their direct, mediating, and total effect on an organization's process innovations. The double-reflective second-order construct model was analyzed following the most recent methodology guidelines. Eventually, partial least squares structural equation modeling (PLS-SEM) was used to test the research hypotheses and investigate the relations between the latent factors. The results from 264 Serbian companies that implemented ISO 9001 standard point to quality management's direct effect on process innovations and knowledge management's mediating effect on process innovation.

**Keywords:** quality management; ISO 9001; standardized management systems; knowledge management; innovation; process innovation; PLS-SEM

## 1. Introduction

Widespread environmental changes force organizations to adapt and innovate their business processes by increasing attention to sustainability and efficiency to remain continuously competitive [1–3]. Quality management and innovations are decisive factors in creating a competitive advantage and improving organizational performance [4–11]. Some researchers proved that the organizational capacity for knowledge management and innovations could determine organizational competitiveness [12].

Over recent years, many studies have attempted to examine the relationship between quality management (QM) and innovation performance [13–17]. Some researchers considered that QM could foster innovation by enabling the efficient detection of customer needs, teamwork, and promoting knowledge sharing, training, commitment, and participation of employees [18–20]. Several studies presented the same results and contended that quality management could be one of the prerequisites for innovation [14–17]. However, some authors claim the opposite [13,18,21–23] and debate the influence of quality management on innovation [22,24,25]. Innovation can be observed as product innovation and process innovation. Many studies confirm a positive QM influence on product innovation. However, there is a gap in research relating to the influence of QM on process innovation (PI) [24]. More importantly, some authors propose that knowledge management (KM) might present the missing link between these two factors [25–29].

Previous empirical research on this topic in the Republic of Serbia showed that quality management partially influences innovation in general, including product and process innovation [30]. Moreover, the authors highlighted that organizational top management shortcomings directly affect innovative performance. They suggested that systematic organizational performance enhancement should include improving managers' commitment to

quality management and knowledge management. However, when attempting to discover if knowledge management indirectly influences innovation, the authors used only one dimension of knowledge management: learning. They concluded that the said relationship is not statistically significant.

Regarding the shortcomings of the presented previous research, this paper aims to investigate knowledge management's importance as a key mediator of a positive influence between quality management and process innovation in Serbia's organizations, including all dimensions of knowledge management. The research instrument was a survey distributed to production and service organizations in Serbia, mainly oriented toward manufacturing, consulting, Information and Communication Technologies (ICTs), education, and public services. The research sample consisted of 264 Serbian companies that implemented ISO 9001 standard [31].

To analyze the survey results and to discover the nature of suspected relationships between QM, KM, and PI, following the most recent methodology guidelines [24] along with the approaches for measurement model evaluation [32–34], the double-reflective second-order construct model was analyzed. The model introduces QM as the second-order factor, which can be indirectly assessed through five sub-factors (leadership, employee management, process approach, customer focus, and continuous improvement). Likewise, the model presents KM as the second-order factor, which can be evaluated indirectly by assessing its three sub-factors (knowledge creation, knowledge application, and knowledge dissemination). These second-order factors, in turn, can also be assessed indirectly by their indicators, which are directly measured. Eventually, partial least squares structural equation modeling (PLS-SEM) was used to test the research hypotheses and investigate the relationships between the latent factors.

Finally, this research contributes to the current state of the art by filling the literature gap in investigating how QM (overall) enhances knowledge management processes and process innovation. To the best of our knowledge, this relationship dynamic between QM, KM, and PI has not been proven before.

The results of this research proved the literature gap mentioned above. Moreover, they could be helpful for organizational managers aiming to enhance process innovations, as it is established that adequately implemented quality management dimensions increase the level of process innovations in organizations. Furthermore, if knowledge management is implemented as well, the overall positive influence of quality management is even higher.

The remainder of this study is structured as follows. The upcoming section describes extant literature, the model structure factors, and hypotheses development. Section 3 presents the methodology and explains the research instrument, including the data collection process, sample demographics, measurement scales, and analysis. Section 4 presents statistical data and results are discussed in Section 5. The paper concludes with practical implications, limitations, and suggestions for future research.

## 2. Theoretical Background

This section compounds the previous research on the relationship between quality management, knowledge management, and process innovations, as well as a taxonomy of the factors determining the quality and knowledge management concepts. These factors are later utilized as building blocks, i.e., second-order constructs of the presented research model.

### 2.1. Quality Management, Knowledge Management, and Process Innovation Factors

The authors previously conducted and published a Systematic Literature Review (SLR) on the principles determining QM, KM, and PI to discover organizations' most commonly applied principles [25]. The said principles influence the operationalization level of QM, KM, and PI in organizations and are later utilized as constructs in a research model.

### 2.1.1. Quality Management

Firstly, when observing QM, the literature review concluded that the principles with the highest frequency are leadership, employee management, process approach, customer focus, and continuous improvement. Therefore, these principles are set as second-order constructs in the research model.

Leadership (L)—Leadership is one of the key elements of QM. The role of management when implementing a total quality environment is pivotal to the success of implementation [35]. The highest ranks of organizational management should set clear goals, mission, vision, and quality politics and apply them in operational processes [28,29,36–38]. Furthermore, the leadership should enable appropriate resources to enhance quality and realize quality goals [28,39]. Thus, the leadership should provide an environment where information regarding quality strategies is readily available and transparent to motivate employees to participate in improving organizational business processes [23,29,36–38]. Leadership at the lower ranks in the organization, such as process owners, should also have a high autonomy level in the decision-making process and take accountability for the results of the said business process [29,36,37,40].

Employee management (EM)—Employees hold a crucial role in fulfilling quality strategies set by leadership. Managers should create a work environment that encourages employees to perform to the best of their abilities [29,41]. Therefore, the employee satisfaction level should be systematically measured and analyzed, and the issues regarding low satisfaction levels should be resolved [28,29,36]. Activities that could improve the employees' satisfaction level and motivate them to improve the quality of business processes in an organization are team building, teamwork, and specialized workshops and training [36]. A well-planned employee training program positively impacts teamwork, minimizes mistakes, and maximizes employee satisfaction [21].

Process approach (PA)—Each defined business process in an organization should be continuously and soundly executed to achieve set goals [42]. Therefore, a QM-oriented organization should manage business processes by identifying, analyzing, and estimating risks and possibilities [38]. In addition, the organization should regularly conduct internal checks and undertake corrective measures in case of systemic non-conformities [36,38,43]. Performance indicators for each process should be defined to evaluate the process's quality and achieve the management function [28,38].

Customer focus (CF)—Customer satisfaction is the final goal of QM. Thus, the organization should regularly measure, analyze, and take action to improve user satisfaction [28,29,36,37]. Customer feedback is an effective method to initiate improvements in organizational processes [29]. Additionally, the organization should determine the user requirements and expectations and consider them when enhancing the quality of existing products/services or designing new products/services [23,28,29,36,37].

Continuous improvement (CI)—The continuous improvement of products, services, and processes is encouraged to be studied to stay competitive in the contemporary industry environment [36]. By implementing continuous improvements, organizations can shorten the production cycle, which has a positive effect on productivity, and, thus, on performance [23]. Organizations should identify processes where it is possible to implement improvements and utilize the results of internal checks for process improvement [36].

### 2.1.2. Knowledge Management

Secondly, when observing KM, the literature review concluded that the factors with the highest frequency are knowledge creation, knowledge application, and knowledge dissemination [25]. Therefore, the research model sets these factors as second-order constructs.

Knowledge creation (KC)—Knowledge creation is a precondition for knowledge management and is a process that should be managed [44]. Top management should establish a work environment that stimulates the development and enhancement of skills, as well as the knowledge required to develop new products/services [29,45]. Furthermore,

new knowledge should be acquired by collecting and analyzing customer and competitor data [28,29].

Knowledge application (KA)—Following the creation of knowledge, the next step is the application of acquired knowledge. Collecting and applying newly acquired knowledge and experiences is crucial for solving newly emerging problems [29,46]. In addition, stakeholder suggestions should be used to improve products, processes, or services [47]. Organizations should be flexible, take advantage of opportunities to improve products/services and processes, and quickly learn, adopt, and implement new latest technological techniques [28,46]. Additionally, organizations that put effort into engaging with external knowledge through a proactive attitude should expect better performance [48].

Knowledge dissemination (KD)—Employees within the organization should exchange knowledge with each other [49]. For this reason, organizations should have procedures and ways to document and disseminate knowledge among employees [28,29]. To disseminate knowledge within the organization, it is crucial to form diverse teams of employees from different parts of the organization to realize certain tasks [47,50]. In addition, organizations should have processes for sharing knowledge with business partners, collecting suggestions from interested parties, analyzing possibilities of using those suggestions, and submitting reports to the managers [29,47].

### 2.1.3. Process Innovation

Process innovation (PI)—To develop process-oriented organizations, the management should discover and reduce activities that do not add value to the production/service processes [29]. In addition, the organization should introduce new work methods into production/service processes and rapidly introduce and adopt innovations in procedures, techniques, and technologies [24,36,37,51]. Successfully implemented process innovations can help organizations gain a competitive advantage by enabling the emergence of new markets and the adoption of innovations previously implemented by other organizations [52].

### 2.2. Related Work

This subsection explains the rationale for the research model and the relationships between research constructs based on previous research empirical results. Analysis of the presented results led to the research hypotheses definition.

Kafetzopoulos et al. examined the extent to which five core dimensions of quality management (leadership and top management support, employee training and involvement, information and learning, process management, and customer focus) are associated as a single factor with product innovation and process innovation. According to the study findings, quality management dimensions that directly contribute to product and process innovation are leadership and top management support, employee training and involvement, information and learning, process management, and customer focus [14].

In the Malaysian manufacturing sector, Yusr et al. determined a positive effect of applying total quality management (TQM) on enhancing knowledge management processes and the relationship between knowledge management and innovation performance. Therefore, proving a well-established TQM (as one set of practices) within the organization leads to a better performance of KM processes and that KM is necessary for achieving the desired innovation performance. Consequently, they suggested that TQM indirectly enhances innovation performance by providing the required predecessor (i.e., knowledge) [28].

Furthermore, Honarpour et al. examined the reciprocal relation between total quality management and knowledge management and their impact on process and product innovation. The data was collected from a survey of 190 research and development unit managers in Malaysia. The results revealed a positive relationship between TQM and KM and TQM and innovation. Additionally, TQM and KM are positively associated with process and product innovation. They concluded that the companies implementing TQM alongside KM could manage their activities efficiently and innovatively [29].

Qasrawi et al. proved the positive impact of total quality management practices in light of leadership, strategic planning, customer focus, teamwork, process management, information, and analysis on organizational performance in Jordanian telecommunications companies. In addition, the mediating effect of knowledge management processes on the relationship between TQM and organizational performance was tested, showing a positive impact [38].

Hamdoun et al. explored the effects of quality management on innovation by addressing the role of knowledge transfer in Tunisia. The results showed that quality management positively influences knowledge transfer and innovation. Moreover, knowledge transfer contributes positively to innovation [27].

Jiménez-Jiménez et al. analyzed the role of total quality management as a precursor of innovation, where knowledge management holds a mediator role in Spain. The research confirmed that management based on TQM helps organizations manage their knowledge better [16].

Delić et al. examined the relationship between quality management and organizational performance in Serbia. The results confirm significant relationships between some dimensions of quality management and organizational performance. However, leadership's impact on customer focus and quality planning, as well as the impact of knowledge management on process management, was not confirmed [30].

Based on the presented previous work and the research goal, the following hypotheses are set:

**H1.** *QM has a positive effect on PI.*

**H2.** *QM has a positive effect on KM.*

**H3.** *KM has a positive effect on PI.*

**H4.** *KM has a positive mediating effect on the relationship between QM and PI.*

### 3. Materials and Methods

*3.1. Measurement*

This research followed a quantitative approach implemented through the survey method. Based on the theoretical models' relevant research review, the conceptual model for the measurement instrument was built by following a second-order construct structure. Consequently, the instrument measured two second-order constructs: quality management and knowledge management, with 10 corresponding first-order constructs based on 42 items. All items were measured on a five-point Likert scale with the meaning of the rating: 1—strongly disagree, 2—disagree, 3—neutral, 4—agree, and 5—strongly agree. The final model with first and second-order constructs and associated manifest variables are presented in Table 1.

**Table 1.** Measurement model constructs with associated manifest variables.

| Construct | Manifest Variable | Sources |
|---|---|---|
| Quality management (QM) | | |
| Leadership (L) | Top management commitment and participation | [28–30,36–38,53] |
| | Employee involvement | [23,29,30,36–38,53] |
| | Acceptance of quality responsibility by top management | [30,36,37] |
| | Empowerment and motivation support | [28–30,53] |
| Employee management (EM) | Employee performance measure, monitoring, and evaluation | [29,30,37] |
| | Development of quality tools and techniques | [28–30,53] |
| | Quality improvement rewards | [29,30,36,37,53] |
| | Existence of quality teams | [28–30,36,53] |
| | Employee satisfaction | [28–30,36] |

**Table 1.** *Cont.*

| Construct | Manifest Variable | Sources |
|---|---|---|
| Process approach (PA) | Preventive action | [30,38] |
| | Corrective action | [30,36,38] |
| | Internal audits | [30,38] |
| | Performance measurement and evaluation | [28,30,36] |
| Customer focus (CF) | Analyzing user opinions and expectations | [23,28–30,36,37] |
| | Customer satisfaction | [28–30,36,37] |
| | Customer relations improvements | [28–30,36,37] |
| Continuous improvement (CI) | Encouraging continuous improvements | [30,38] |
| | Identification of areas suitable for improvements | [30,36] |
| | Time-based process efficiency | [23,30] |
| | Reduction of unnecessary expenses in processes | [23,30] |
| | Quality improvements through specific organizational structures | [30,36] |
| Knowledge management (KM) | | |
| Knowledge creation (KC) | Generating new knowledge from existing knowledge | [28,29,54] |
| | Acquiring knowledge about new products within a specific industry | [28,29,47] |
| | Capturing knowledge of our competitors | [28,29,47] |
| | Employee training | [29,47] |
| Knowledge application (KA) | Responds quickly to changing technology | [28,38] |
| | Responds quickly to changing products, processes, and strategies | [28,38] |
| | Applying knowledge to solve new problems | [28,29,38] |
| Knowledge dissemination (KD) | Distributing knowledge throughout the organization | [28,29,54] |
| | Distributing knowledge among business partners | [29,54] |
| | Teamwork | [47,54] |
| Process innovation (PI) | | |
| Process innovation (PI) | Determining and eliminating non-value-adding activities in production processes | [29,53] |
| | Introducing new methods for the production process | [23,36,37,53] |
| | The rate of change in processes, techniques, and technology | [23,36,37,53] |
| | The speed of adopting the latest technological innovations in processes | [23,36,37] |

### 3.2. Data Collection

Before the research procedure started, a pilot study was conducted to identify potential ambiguities and unclarities related to the measuring instrument. A pilot study was conducted on 30 competent respondents' samples, including quality managers or directors of organizations from the production and service sectors.

After the final version of the questionnaire was created, the quantitative data collection lasted over six months, from May to November 2022. The questionnaire was distributed via e-mail to the Republic of Serbia's organizations that have implemented the ISO 9001 standard.

Respondents were contacted by phone first, requesting consent to participate in the study. Afterwards, according to Dillman's adapted approach, the questionnaire distribution was accompanied by a series of reminders to complete the survey and increase the respondents' response rate [55]. Respondents accessed the online survey via Survey Monkey. Of the 400 respondents contacted, the final sample size is 264. Demographics of the sample, including gender, age, number of employees, type of organization, organization category, and work experience in the field, are shown in Table 2.

**Table 2.** Sample demographics.

| Variable | Classification | N (Frequency) | % (Percent) |
|---|---|---|---|
| Gender | Male | 135 | 51.1% |
| | Female | 129 | 48.9% |
| Age | Less than 30 | 110 | 41.7% |
| | Between 31 and 40 | 66 | 25.0% |
| | Between 41 and 50 | 48 | 18.2% |
| | More than 51 | 40 | 15.1% |
| Number of employees | Between 1 and 10 | 35 | 13.3% |
| | Between 11 and 49 | 45 | 17.0% |
| | Between 50 and 249 | 65 | 24.6% |
| | More than 250 | 119 | 45.1% |
| Type of organization | Production | 74 | 28% |
| | Service | 121 | 45.8% |
| | Production and service | 69 | 26.2% |
| Organization category | Manufacturing | 116 | 43.9% |
| | Consulting | 22 | 8.3% |
| | Information and Communication Technologies (ICTs) | 32 | 12.1% |
| | Education | 16 | 6.1% |
| | Public services | 11 | 4.2% |
| | Mining and energetics | 22 | 8.3% |
| | Banking | 6 | 2.3% |
| | Health care | 7 | 2.7% |
| | Agriculture | 8 | 3% |
| | Chemical and pharmaceutical | 6 | 2.3% |
| | Other | 18 | 6.8% |
| Work experience in the field | Less than 10 | 166 | 62.9% |
| | Between 11 and 20 | 60 | 22.7% |
| | Between 21 and 30 | 28 | 10.6% |
| | More than 31 | 10 | 3.8% |

*3.3. Analysis Method*

This research used partial least squares structural equation modeling (PLS-SEM) to identify relationships between constructs, focusing on explaining the variance in the dependent variables when examining the model [56].

Magno et al. guided the PLS-SEM method's utilization in quality management [57]. The double-reflective second-order construct measurement model was analyzed following these guidelines and guidelines for measurement model evaluation [32–34,56]. Statistical analysis was performed using SmartPLS 4 software. The statistical analysis took the three stages approach. To begin with, we validated the first-order measurement model, and afterwards, we validated the second-order measurement model. At long last, we tested the structural model. The repeated indicators approach was utilized since it produces smaller biases in assessing the higher-order measurement model. Latent variable scores are used when looking at the structural model [32,56].

**4. Results**

Collected data were analyzed to validate the measurement instrument. The Cronbach alpha coefficient α was used to validate the instrument's reliability and validity. Table 3 shows the Cronbach alpha coefficient for all factors and the instrument. Furthermore, the measurement model testing was approached following three steps explained below.

**Table 3.** First-order construct reliability and convergent validity.

| Construct | Items | Outer Loadings | α | CR | AVE |
|---|---|---|---|---|---|
| Leadership | L2 | 0.785 | 0.798 | 0.797 | 0.568 |
| | L3 | 0.728 | | | |
| | L5 | 0.745 | | | |
| Employee Management | EM1 | 0.741 | 0.844 | 0.845 | 0.577 |
| | EM3 | 0.722 | | | |
| | EM4 | 0.777 | | | |
| | EM5 | 0.796 | | | |
| Process Approach | PA1 | 0.772 | 0.849 | 0.850 | 0.588 |
| | PA2 | 0.745 | | | |
| | PA4 | 0.703 | | | |
| | PA5 | 0.841 | | | |
| Customer Focus | CF2 | 0.908 | 0.812 | 0.813 | 0.596 |
| | CF3 | 0.661 * | | | |
| | CF4 | 0.725 | | | |
| Continuous Improvement | CI1 | 0.695 * | 0.863 | 0.862 | 0.611 |
| | CI2 | 0.754 | | | |
| | CI3 | 0.824 | | | |
| | CI4 | 0.844 | | | |
| Knowledge Creation | KC2 | 0.802 | 0.779 | 0.780 | 0.544 |
| | KC3 | 0.651 * | | | |
| | KC4 | 0.768 | | | |
| Knowledge Application | KA1 | 0.719 | 0.858 | 0.858 | 0.602 |
| | KA2 | 0.769 | | | |
| | KA3 | 0.814 | | | |
| | KA4 | 0.797 | | | |
| Knowledge Dissemination | KD1 | 0.881 | 0.824 | 0.821 | 0.538 |
| | KD3 | 0.712 | | | |
| | KD4 | 0.652 * | | | |
| | KD5 | 0.675 * | | | |
| Process Innovation | PI1 | 0.741 | 0.860 | 0.859 | 0.550 |
| | PI2 | 0.812 | | | |
| | PI3 | 0.718 | | | |
| | PI4 | 0.703 | | | |
| | PI5 | 0.730 | | | |
| Whole instrument | | | 0.941 | | |

* These indicators were kept in the measurement model even if they did not hit the threshold since no effect on CR increase was found [56].

### 4.1. Assessment of First-Order Measurement Model

The validity of the first-order measurement model was evaluated by examining the convergent and discriminant validity of all first-order reflective factors. Convergent validity was established through the outer loadings, Cronbach's alpha, composite reliability (CR), and average variance extracted (AVE) [58].

First, factor analysis was done iteratively until an adequate model and factor structure that satisfies all the criteria was achieved. Factor loadings can range from $-1.0$ to $+1.0$, with higher absolute values indicating a higher correlation of the item with the underlying factor [59]. Among 42 items that were analyzed, eight were omitted according to the recommendations of Hair et al. (2017). Their outer loadings were between 0.40 to 0.70; thus, deleting these indicators increased CR and AVE [56].

Second, Cronbach's Alpha ranged from 0.779 to 0.863, whereas composite reliability statistics ranged from 0.780 to 0.862. Both indicators have reliability statistics over the

required threshold of 0.70 [56]. Moreover, observing composite reliability, all values are between 0.70 and 0.90 and can be regarded as satisfactory [56]. Hence, construct reliability is established.

Third, AVE was used to establish convergent validity on the construct level. All AVE values are above 0.5, which indicates that the construct explains more than half of the variance of its indicators [56]. All results are presented in Table 3.

In addition to examining the HTMT ratios, the second recommendation is to test whether the HTMT values differ significantly from 1 [56]. We calculated the confidence intervals by running the 5000 bootstrap samples. In the columns marked with 2.50% and 97.50%, the lower and upper limits of the 95% confidence interval are presented (with bias-corrected and accelerated options applied). It is observed that no interval includes the value 1, which is a criterion for establishing discriminative validity. HTMT is the recommended method for discriminating validity evaluation in PLS-SEM. Therefore, based on these results, the discriminative validity of the constructs was determined (Table 4).

**Table 4.** HTMT confidence interval for first-order constructs.

| | Original Sample (HTMT) | 2.5% | 97.5% |
|---|---|---|---|
| L -> PI | 0.729 | 0.621 | 0.824 |
| PA -> PI | 0.677 | 0.567 | 0.768 |
| PA -> L | 0.853 | 0.777 | 0.917 |
| EM -> PI | 0.755 | 0.673 | 0.828 |
| EM -> L | 0.868 | 0.795 | 0.933 |
| EM -> PA | 0.838 | 0.764 | 0.907 |
| CF -> PI | 0.647 | 0.524 | 0.754 |
| CF -> L | 0.700 | 0.573 | 0.807 |
| CF -> PA | 0.867 | 0.791 | 0.913 |
| CF -> EM | 0.706 | 0.606 | 0.792 |
| KA -> PI | 0.904 | 0.847 | 0.954 |
| KA -> L | 0.791 | 0.701 | 0.867 |
| KA -> PA | 0.761 | 0.672 | 0.840 |
| KA -> EM | 0.756 | 0.659 | 0.837 |
| KA -> CF | 0.670 | 0.550 | 0.769 |
| KD -> PI | 0.641 | 0.518 | 0.755 |
| KD -> L | 0.754 | 0.653 | 0.839 |
| KD -> PA | 0.790 | 0.712 | 0.861 |
| KD -> EM | 0.765 | 0.660 | 0.858 |
| KD -> CF | 0.757 | 0.652 | 0.847 |
| KD -> KA | 0.810 | 0.711 | 0.893 |
| CI -> PI | 0.722 | 0.627 | 0.800 |
| CI -> L | 0.807 | 0.724 | 0.884 |
| CI -> PA | 0.856 | 0.791 | 0.929 |
| CI -> EM | 0.789 | 0.696 | 0.870 |
| CI -> CF | 0.842 | 0.756 | 0.915 |
| CI -> KA | 0.744 | 0.654 | 0.819 |
| CI -> KD | 0.819 | 0.745 | 0.888 |
| KC -> PI | 0.728 | 0.609 | 0.834 |
| KC -> L | 0.865 | 0.763 | 0.949 |
| KC -> PA | 0.848 | 0.765 | 0.922 |
| KC -> EM | 0.869 | 0.783 | 0.944 |
| KC -> CF | 0.736 | 0.615 | 0.845 |
| KC -> KA | 0.823 | 0.732 | 0.900 |
| KC -> KD | 0.835 | 0.733 | 0.923 |
| KD -> CI | 0.762 | 0.647 | 0.857 |

### 4.2. Assessment of Second-Order Measurement Model

The second-order measurement model was validated following the steps recommended by Becker et al., Hair et al., Magno et al., and Sarstedt et al. [32,34,57,60].

The first second-order construct, quality management, was based on five first-order constructs: leadership, employee management, process approach, customer focus, and continuous improvement. The second second-order construct, knowledge management, was based on three first-order constructs: knowledge creation, knowledge application, and knowledge dissemination. QM and KM are measured as reflective–reflective second-order constructs in the study.

The second-order constructs were tested for reliability and convergent and discriminant validity belonging to first-order constructs.

Reliability was assessed using Cronbach's alpha ($\alpha$) and composite reliability (CR), and all the values for reliability were more significant than the recommended value of greater than 0.70. Convergent validity was acceptable because all AVE values are more effective than 0.50 (Table 5). The results for the second-order constructs' reliability and validity appear to be established.

**Table 5.** Second-order construct reliability and convergent validity.

| Construct | $\alpha$ | CR | AVE |
|---|---|---|---|
| QM | 0.920 | 0.940 | 0.759 |
| KM | 0.865 | 0.917 | 0.787 |

Discriminant validity of the second-order constructs with the lower-order constructs was also evaluated through HTMT criteria. Likewise, in first-order constructs, the HTMT for second-order constructs was calculated by the HTMT values differentiation from 1 testing. Hence, no interval includes the value 1, and based on that, the construct's discriminant validity was presented. These results are shown in Table 6.

**Table 6.** HTMT confidence interval for second-order constructs.

| Construct | Original Sample (HTMT) | 2.5% | 97.5% |
|---|---|---|---|
| QM <-> PI | 0.775 | 0.690 | 0.840 |
| KM <-> PI | 0.834 | 0.744 | 0.907 |
| KM <-> QM | 0.941 | 0.887 | 0.981 |

*4.3. Structural Model*

Since the measurement model assessment indicated satisfactory quality, we proceeded with the structural model testing.

Standard assessment criteria, which should be considered, include the variance inflation factor (VIF), the coefficient of determination ($R^2$), the cross-validated redundancy measure $Q^2$, and the statistical significance and relevance of the path coefficients [56,61]. We used the Smart PLS algorithm to calculate path coefficients and determine the value of $R^2$. Bootstrapping was employed to test for statistical significance and draw at least 10,000 bootstrap samples [57,60]. In addition, the model's out-of-sample predictive power was assessed using the PLSpredict procedure [56].

At first, the variance inflation factor of all sets of predictor constructs in the structural model was tested [56].

VIF—We concluded that there is no indication of collinearity between each set of predictor variables since all VIF values, QM -> I = 3.386, KM -> I = 3.386, and QM -> KM = 1.000, are below the threshold of 5 [56].

$R^2$—The $R^2$ value ranges from 0.551 to 0.705, with higher values indicating higher levels of predictive accuracy. The overall $R^2$ is moderate, suggesting that the two constructs, QM and KM, can jointly explain 55.1% of the variance of the endogenous construct IP. Another finding to QM explains 70.5% of KM'S variances in this model, which represented a strong coefficient of determination.

$Q^2$—According to [62,63], in addition to evaluating the $R^2$ values, we also examined Stone-Geisser's $Q^2$ value. The $Q^2$ value is used to assess the predictive power of the structural model, i.e., to judge the model's predictive relevance concerning each endogenous construct [56]. To calculate the $Q^2$ value, the PLSpredict algorithm was used [32]. Hair et al. recommended that $Q^2$ values greater than 0 confirm the predictive validity of the structural model [32]. For the construct KM, the $Q^2$ value is 0.703, and for PI, it is 0.474. Therefore, since these values are greater than 0, they indicate good predictive relevance regarding the endogenous latent variables. Finally, the $Q^2$ values for the endogenous constructs were over 0. Hence, predictive relevance was established.

Path analysis—The following step in structural equation modeling assesses the hypothesized relationship to substantiate the proposed hypotheses. The hypotheses were examined with path analysis (Figure 1).

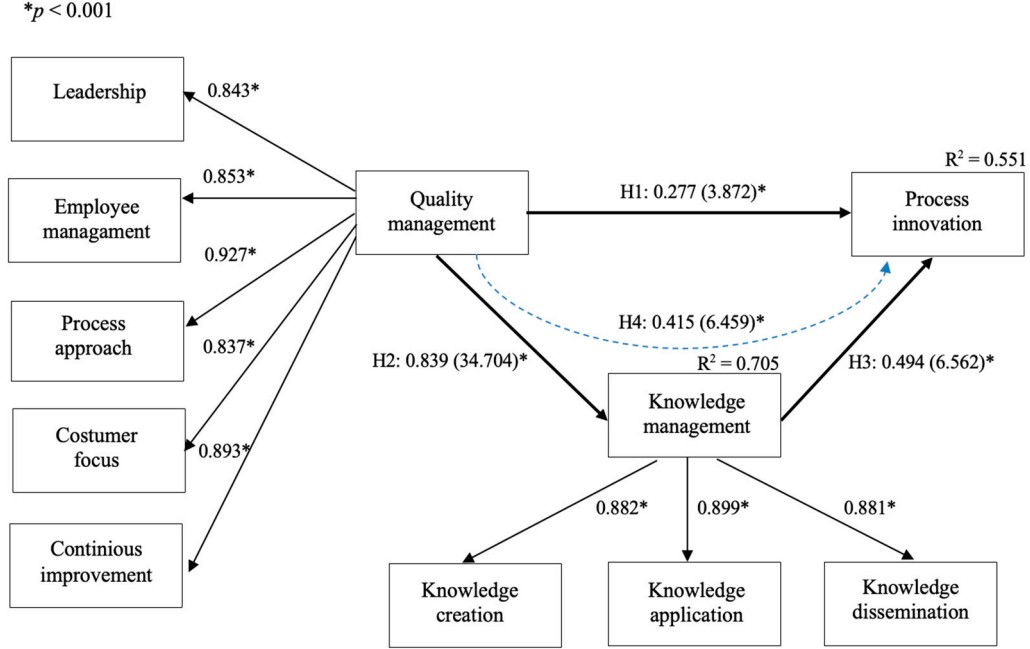

**Figure 1.** SEM model.

The values of the path coefficients confirmed the assumed relationships between the factors in the model because all examined direct and indirect effects in the structural model are statistically significant. If we look at the relationship's strengths between the constructs, the most substantial effect was obtained for path quality management -> knowledge management (H2: $\beta$ = 0.839, t = 34.704, $p < 0.001$). An impact of somewhat weaker intensity was obtained for path KM -> PI (H3: $\beta$ = 0.494, t = 6.562, $p < 0.001$), and the effect of the weakest intensity was obtained for path QM -> PI (H1: $\beta$ = 0.277, t = 3.872, $p < 0.001$).

Mediation analysis was performed to assess the specific indirect effect of QM. The results revealed a significant ($p < 0.001$) mediating role of KM (H4: $\beta$ = 0.415, t = 6.459, $p < 0.001$).

Nevertheless, besides evaluating the construct's direct effect on another, an assessment of its indirect effects via mediating construct has also been made. The sum of direct and indirect effects is called the total effect [56]. Although the direct effect of QM on PI is slightly solid ($\beta$ = 0.277, t = 3.872, $p < 0.001$), the total effect is quite pronounced ($\beta$ = 0.692, t = 18.441, $p < 0.001$), indicating the relevance of QM in explaining PI. These results suggest that KM mediates the relationship between QM and PI. The above analysis shows that all assumed hypotheses (H1–H4) were supported.

## 5. Discussion

In organizations, PI, KM, and QM have received significant attention from academics worldwide, providing different insights [64]. In the last decade, studies investigating the importance of QM and KM and their impact on the organization's performance proved that QM and KM are key factors for continuously improving processes [26–29]. In addition, during the last few years, improving processes have been increasingly reflected in the organization's ability to apply innovations in all fields of business [24]. Finally, several studies examined the relationship between QM, KM, and the organization's innovative capability. Based on a comprehensive literature review and statements presented above, this study was motivated to understand better the importance of the QM and KM existence and their direct and mediating effect on an organization's PI.

Accordingly, we extend the current literature on processes innovations predictors by proposing the research model where the relationship between QM, KM, and processes innovations is demonstrated, along with QM's mediating effect on PI by KM.

This study offers a reliable and valid second-order model demonstrating the relationship between QM, KM, and PI. The model introduces QM as the second-order factor, which can be indirectly assessed through five sub-factors (leadership, employee management, process approach, customer focus, and continuous improvement). Additionally, the model presents KM as the second-order factor, which can be evaluated indirectly by assessing its three sub-factors (knowledge creation, knowledge application, and knowledge dissemination). These second-order factors, in turn, can also be indirectly evaluated by their indicators, which are directly measured.

To prove the questioned relationships proposed through the model from this study, we followed the PLS-SEM utilization methods in QM [32]. Hence, the double-reflective second-order model was analyzed after the measurement model evaluation.

The quality managers' perceptions in the Republic of Serbia companies have been used for testing the hypotheses and the model presented in the study.

The first-order measurement model met the criterion for convergent and discriminant validity (see Table 3). The reflective second-order measurement model met the reliability and validity requirements (see Table 5). Discriminant validity for the first-order factor of process innovation and second-order factors of quality management and knowledge management was established (see Tables 4 and 6).

We found that all relationships in the structural model were statistically significant, where the relationship between QM and KM ($\beta = 0.839$, $p < 0.001$) was seen as the most important, followed by the relationship between KM and PI ($\beta = 0.494$, $p < 0.001$). Lastly, the relationship between QM and PI ($\beta = 0.277$, $p < 0.001$) was found to be the least impactful. However, the total effects analysis results suggested that KM mediates the relationship between QM and PI ($\beta = 0.415$, $p < 0.001$).

This study supports organizations implementing quality and knowledge management to develop process innovations. Organizations that consider their process innovation development inadequate but already apply quality management should include knowledge management, thus raising innovation development to a greater extent.

Therefore, the positive results of this study regarding the relationship between QM and KM, QM and PI, and KM and PI reinforce the corresponding earlier findings in the literature [26–29].

Another significant contribution of the study is the total effects analysis results that suggest KM mediates the relationship between QM and PI ($\beta = 0.692$, $p < 0.001$).

The awareness of QM existence in the organizations is shown as very high and impactful, looking through all first-order factors from this study. Leadership ($\beta = 0.843$, $p < 0.001$), employee management ($\beta = 0.853$, $p < 0.001$), process approach ($\beta = 0.927$, $p < 0.001$), costumer focus ($\beta = 0.837$, $p < 0.001$), and continuous improvement ($\beta = 0.893$, $p < 0.001$) strongly support the importance of QM in organizational structure.

On the other hand, observing the first-order factors from KM, it is proven that they contribute to KM composition with the following impacts: knowledge creation ($\beta = 0.882$,

$p < 0.001$), knowledge application ($\beta = 0.899$, $p < 0.001$), and knowledge dissemination ($\beta = 0.881$, $p < 0.001$).

*Managerial Implications*

The presented study's empirical findings generally emphasize QM and KM's importance in enabling process innovation and providing a competitive advantage. This study's contribution can be observed through the theoretical and practical implications explained below.

This study's positive results support organizations implementing the ISO 9001 standard to develop process innovations. In addition, they also motivate managers to improve their quality management system if it does not give them results, putting a particular focus on KM development.

Guidelines for implementing standards or QMS improvement are based on the adequate application of all QM practices: leadership, employee management, process approach, customer focus, and continuous improvement. It starts with emphasizing the leadership commitment critical to implementing and improving QM. Leadership should set clear goals, mission, vision, and strategies focused on process innovation and providing resources to achieve them. Furthermore, motivating and involving employees in developing and implementing business process improvements is necessary. Given that QM is based on a process approach, lower-level managers should also have high autonomy. It should include a decision-making process, assessing risks and opportunities, defining key performance indicators, and improving the mentioned business process results.

Previous researchers have considered that managers should enhance applying TQM practices at the strategic level of the organizations, which means giving more attention to previously mentioned practices [27,28,38].

When an organizational culture empowers employees to be free and motivated to make suggestions for innovation, it is possible to focus on implementing other QM practices that will contribute to performance, i.e., customer orientation. Nonetheless, organizations should consider their users' expectations and requirements when improving the quality of existing products/services or designing new products/services. Additionally, customer feedback is an effective method for improving organizational processes. In addition, as a basis for continuous improvements in the organization, it is necessary to use the results of previously conducted internal checks, i.e., system reviews. The aforementioned information enables employees to discover and reduce activities that do not add value to the production/service processes. It also contributes to defining new work methods, procedures, techniques, and technologies in production/service processes. Consequently, managers should strongly emphasize these QM practices, as they are essential for achieving process innovation and sustainable competitive advantage. Additionally, they will perform with even better results in developing process innovations if they apply KM practices.

KM practices include acquiring, applying, and disseminating employee knowledge to raise employee awareness to develop process innovations.

The study findings of Yusr et al. also indicate that KM process enhancement demands TQM practices [28]. In that way, practices can complement each other and ensure the flow and update of cutting-edge knowledge throughout the organization [28]. In addition, Honarpour et al. suggest that TQM should play a pivotal role in KM development [29]. KM should have an increased chance of success through a TQM focus and the effective use of knowledge towards innovation [29]. Accordingly, Qasrawi et al. emphasized that top management should set an example for their employees by sharing their knowledge and establishing effective mechanisms for knowledge application and sharing [38]. In this sense, Jiménez-Jiménez et al. recommended that managers implement tools and techniques, such as brainstorming, taxonomy, COP, social networking services, or advanced search tools, to spread knowledge in an organization [16].

Generally observed, top management is responsible for creating a working environment that stimulates the development and improvement of knowledge and skills. This

paper identified some mechanisms that recommend employees acquire knowledge through daily work, solve various problems, cooperate with employees, and attend specific training.

As claimed by Honarpour et al., KM practices help to identify points of quality improvement and ensure that innovation will be of the utmost importance in guiding organizational responses to market changes [29]. Therewith, by constantly examining the market, employees acquire new knowledge and apply it in developing new ideas, suggestions for improvements, and quickly changing products, processes, strategies, and technologies.

To conclude, with the adequate application of QM and KM practices utilized by all employees, the process innovation development will rise to a high level. Ultimately, this will ensure that organizations maintain a competitive advantage through contoured and continuous improvements.

## 6. Conclusions

This study focuses on the relationship between QM, KM, and PI, studying the effect of five QM sub-factors and three KM sub-factors on PI. The proposed model simultaneously considers the relationships between QM and KM and their total, direct, and indirect influence on PI.

Observing the previous research from the Republic of Serbia, the lack of a statistically significant, direct relationship between leadership and employee management on the one hand and customer focus on the other may point to the shortcomings of top managers' work [30]. Altogether, this brought us to examine these QM items' effect on PI.

This study attempted to investigate how effective QM is in enhancing KM processes and PI and clarify the role of KM processes in improving PI. Thus, the first contribution of the presented study is the close examination and better understanding of these relationships.

Our research showed that 60% of respondents confirmed that top management information sharing with employees at meetings and including them in the discussions of critical quality issues led to innovation. Moreover, 70% of the respondents claimed that top management motivates and encourages employees to participate in proposals aimed at innovation. Finally, this concludes that leadership dedication and employee inclusion have risen in the past few years. However, it is still necessary to focus on it.

In addition, CF is at an extremely high level. More than 75% of respondents claimed that this factor is essential; therefore, it confirmed that companies regularly measure, investigate, and fix customer satisfaction and have a practical approach to analyzing customer information. Furthermore, current processes are modified and enhanced based on customer feedback, which implies the existence of process innovations. These responses demonstrate a good practice that all organizations should follow.

When a direct effect of QM on PI is observed, there is a positive relationship. However, the said relationship is weaker than the total effect on PI when the KM effect on PI is included. The addressed relationship illustrates the importance of the KM's existence in the organization's ecosystems where QM already affects PI.

The results of this research can help organizations employ quality management and knowledge management to achieve process innovations as the ultimate outcome.

This research has some limitations that should be highlighted. First, this study has only examined the relationships between QM and KM and their total, direct, and indirect influence on an organization's PI. Therefore, it is recommended to examine the moderating role of KM to provide more sight regarding this issue. Second, this study considers only PI, but future studies might consider product innovation. Third, this study was limited to companies in the Republic of Serbia. Consequently, retesting the model in different countries will enhance the generalizability of the gained results in this study.

Eventually, in future research, it would be valuable to conduct a longitudinal study to detect the influence of QM and KM on PI throughout time.

**Author Contributions:** Conceptualization, M.Ž., T.V. and M.D.; Methodology, M.Ž., T.V. and M.D.; Software, T.V.; Validation, M.Ž. and M.D.; Investigation, M.Ž.; Resources, S.V.; Data curation, M.Ž.; Writing—original draft, M.Ž., T.V. and D.D.; Writing—review & editing, M.Ž., T.V. and D.D.; Visualization, M.Ž., S.V. and M.D.; Supervision, T.V., S.V. and D.D. All authors have read and agreed to the published version of the manuscript.

**Funding:** This research received no external funding.

**Institutional Review Board Statement:** Not applicable.

**Informed Consent Statement:** Not applicable.

**Data Availability Statement:** Not applicable.

**Acknowledgments:** The authors thank the anonymous reviewers for their valuable comments and suggestions.

**Conflicts of Interest:** The authors declare no conflict of interest.

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
