# Peer review of "Investigating the Key Factors Influencing the Process Innovation Capability in Organizations: Evidence from the Republic of Serbia"

_sustainability, doi:10.3390/su15108158_

Round 1

Reviewer 1 Report

I highly recommend that the authors revisit the logical standing of the introduction and the quality of communication of the manuscript on page 1,2

 Results are presented clearly and analyzed appropriately and the conclusions adequately tie together the other elements of the paper

The paper's argument is built on an appropriate base of theory and concepts

Reviewer 2 Report

1.       What kind of organization this study means? The use of the word organization here must be clarified and specified. The word organization is too broad.

2.       This article is lack of information about the background of the study

3.       Eventhough the authors claimed that this research contributes to the current state of the art by filling the literature 50 gap in investigating how QM (overall) enhances knowledge management processes and 51 process innovation. I do not think this study introduced something new discussion.

4.       Is there any validity and reliability test for the data?

5.       This study only focus on SEM result, buat lack of scientific discussion that friendly for the reader

6.       The flow of writing is too pushy

Reviewer 3 Report

The paper has several strengths that make it a valuable contribution to the academic community. Firstly, the study focuses on a critical area of research that has received significant attention from scholars worldwide. Secondly, the paper highlights the importance of quality management and knowledge management and their impact on an organization's process innovations. Thirdly, the study uses a robust and reliable research methodology, including the double-reflective second-order construct model and partial least squares structural equation modeling (PLS-SEM) to test the research hypotheses and investigate the relations between the latent factors.

However, there are a few points which need the authors attention in order to make this paper a great addition to the journal:

1. Based on the findings of this research, it is recommended to create a managerial implications paragraph that highlights the potential benefits of employing quality management and knowledge management practices in organizations. Here, the authors should discuss practical implications and recommendations for organizations based on the study's findings.

2. I would suggest that the following articles can be used to improve the paper's literature review on competitiveness, innovation, and leadership:

https://doi.org/10.3390/logistics7010013  

These articles are highly relevant to the topic of the paper and provide valuable insights into the relationship between competitiveness, innovation, and leadership. Incorporating these articles into the literature review can help the paper to establish a more solid theoretical foundation and enhance the overall quality of the research. 

It is recommended to check English consistency and grammar from the abstract and throughout the paper.

Round 2

Reviewer 2 Report

Thank you in advance for giving me opportunity to review this revised version. The authors have made significant improvements. However, there are comments that are still unaddressed. 

1.       There is some repetition on introduction (See paragraph 5 and 6). Please clarify the introduction.

2.       Please write the survey year of the data used in this study. When data collection was done?

3.       Strengthen your discussion part in view of previous relevant or global studies. Especially on discussion sub section 5.1.

4.       Please check minor errors in punctuations, in-text citations, appropriateness of subtitles, and formatting (placements of tables/figures relative to text).

Minor editing of English language required
